# Replacing Toxic Hard Chrome Coatings: Exploring the Tribocorrosion Behaviour of Electroless Nickel-Boron Coatings

Muslum Yunacti [1,*], Veronique Vitry [1,*] , Alex Montagne [2] and Mariana Henriette Staia [3,4]

1 Service de Métallurgie, Faculté Polytechnique, Université de Mons, Rue de l'Epargne, 56, 7000 Mons, Belgium
2 Campus Mont Houy, LamiH UMR 8201 CNRS, Université Polytechnique Hauts-de-France, 59313 Valenciennes, France; alex.montagne@uphf.fr
3 School of Metallurgical Engineering and Material Science, Faculty of Engineering, Universidad Central de Venezuela, Los Chaguaramos, Caracas 1040, Venezuela; mhstaia@gmail.com
4 Venezuelan National Academy for Engineering and Habitat, Palacio de las Academias, Caracas 1010, Venezuela
* Correspondence: muslum.yunacti@umons.ac.be (M.Y.); veronique.vitry@umons.ac.be (V.V.)

**Abstract:** Electroless nickel-boron coatings present outstanding properties such as high hardness, excellent wear resistance and uniform coating, and thus they are considered to be alternative to toxic hard chrome coatings. However, they contain lead that is toxic and used as stabilizer in the plating bath. This study aims to investigate the tribocorrosion behaviour of lead-free electroless nickel-boron coatings. In the present research, several tests were carried out to investigate the behaviour of these coatings under both dry and tribocorrosion reciprocating sliding wear against alumina balls, at room temperature. The open circuit potential (OCP) method was used to determine the degradation mechanism of the coatings. The results of the tribocorrosion and dry wear tests showed that the performance of coatings was very different from each other. A steady state for the coefficient of friction (COF) is achieved during the tribocorrosion test, whereas the constant production of debris and their presence in the contact implied an increase in COF with distance during the dry wear test. The wear mechanisms of these coatings also presented variations in these tests. It was found that the wear area calculated from tribocorrosion is lower ($56~\mu m^2$) than the one from dry sliding test ($86~\mu m^2$).

**Keywords:** electroless nickel-boron coatings; tribocorrosion; open circuit potential; friction coefficient; wear mechanism

## 1. Introduction

Electroless nickel plating has gained significant attention from researchers owing to several beneficial characteristics: low-cost processing, ability to deposit uniform coatings on complex shaped substrates, ability to deposit on electrically non-conductive substrates, high hardness coatings, low friction coefficient (COF), improved corrosion and wear resistance [1,2]. This process is an autocatalytic chemical reduction process that involves the reduction of nickel ions in an aqueous solution by oxidation of a reducing agent [3]. Different reductants, hypophosphite [4], borohydride [5], amine borane [6] and hydrazine, are used for this process [7]. Depending on the reducing agent, nickel alloys (ENB or ENP) or pure nickel deposits are obtained [7–9].

Electroless nickel-boron (ENB) deposits are obtained when a borohydride reducing agent is used. These deposits are the hardest and the most wear resistant among all electroless nickel coatings [10–12]. The properties of these deposits mainly depend on the boron content. These deposits have been used to protect and enhance various substrates such as steel, stainless steel and titanium, which are extensively used in several industries [13,14].

Materials plated with ENB can be exposed to wear and corrosion at the same time, and synergy can result in higher degradation rates than their individual contributions [15,16].

In order to investigate the simultaneous effects of corrosion and wear, tribocorrosion tests need to be carried out in a tribometer integrated with a potentiostat, able to perform electrochemical measurements during the immersion of the triboelements in the corrosive solution.

Many researchers have carried out studies on the resistance of electroless nickel coatings to wear or corrosive attack separately [17–20]. However, detailed studies about the synergy between wear and corrosion of electroless coatings are scarce [14,15,21]. As it was indicated recently by Siddaiah et al. [22] many researchers follow the OCP measurement along with sliding tests to evaluate the tribocorrosion performance of metallic components. While measuring the tribocorrosion at OCP, a stable OCP is first achieved, which is followed by sliding against a suitable counterpart. During sliding, a shift in OCP can be observed based on whether the metal system is active or passive. Salicio-Paz et al. [15] showed, for monolayered and multilayered ENP electroless coatings submitted to tribocorrosion tests in a 3.5 wt.% NaCl solution, that multilayer coatings cope better with the stress derived from the applied normal load and sliding motion, which translates into a lower plastic deformation of the coatings and 3.5 times lower worn volumes under the same testing conditions. They also showed that ENP coatings do not show passivation in 3.5 wt.% NaCl solution.

Vitry and Bonin [21] performed tribocorrosion tests for ultrasonic assisted electroless nickel-boron deposition, in a modified pin-on-disc tribometer using a normal load of 5 N, speed 5 cm/s, total length 100 m and 0.1% NaCl corrosive solution. They determined that the samples with a thickness of 25 μm synthesized at 95 °C with ultrasound assistance exhibited a very low specific wear rate, small friction coefficient and no shift in the open circuit potential (OCP) value during the tribocorrosion tests.

ENB coatings are produced in a bath containing different components and one of them is stabilizer. Generally, heavy metal stabilizers like lead are used and their uses are limited due to their toxicity. Recently several lead-free ENB coatings have been developed [23–25].

The research group from Mons University, Belgium, has recently published results related to the production, mechanical and corrosion resistance properties of a novel ENB electroless coatings produced in a stabilizer-free bath that does not contain lead [26]. These novel ENB coatings are essential for the assessment of their suitability as a replacement for toxic hard chrome plating. As mentioned above, ENB coatings are used in applications where they are exposed to wear and corrosion at the same time. No research has been found that studies tribocorrosion behaviour of lead-free ENB coatings. Therefore, the present research investigates the behavior of these lead-free ENB coatings with the aim of determining their performance when the combined attack of reciprocating sliding wear and corrosion occurs simultaneously. In this research, novel ENB coatings were tested in the tribocorrosion test system to examine COF and OCP. Additionally, reciprocating dry sliding wear test was also carried out for performance and mechanism of wear comparison. After the tests, the coatings were analyzed by means of SEM and optical profilometer in order to investigate wear mechanisms and wear area.

## 2. Materials and Methods

### 2.1. Substrate Preparation

The substrate used in these experiments was a 1 mm thick ST 37-DIN 17100 mild steel plate, that was cut to size according to the bath loading (25 cm$^2$/L). SiC papers of 180, 500 and 1200 grit were used for their metallographic preparation. Subsequently, they were cleaned in acetone prior to their activation in a solution of 30% vol. HCl for 3 min. After each step, the substrates were rinsed with distilled water.

### 2.2. Electroless Nickel-Boron Plating

The composition of the electroless nickel-boron plating bath is shown in Table 1. The deposition was carried out on a regulated hot plate with magnetic stirring. The volume of the plating bath was one liter (1 L). Process temperature was fixed at 95 ± 1 °C and

its duration was one hour. In these experimental conditions, a nickel-boron deposit with a thickness of 15 μm (determined through cross-section) was obtained with a uniform surface, a featureless and dense cross-section. The deposit had approximately 4 wt.% B and 96 wt.% Ni and was not fully amorphous but contained significant short range order areas [27].

**Table 1.** Composition of electroless nickel-boron plating bath.

| Compound | Amount [16] |
|---|---|
| NiCl$_2$·6H$_2$O (99%—VWR Chemicals) (g/L) | 24 |
| NaBH$_4$ (99.9%—Acros Organics) (g/L) | 0.4 |
| NH$_2$-CH$_2$-CH$_2$-NH$_2$ (99% VWR Chemicals) (mL/L) | 120 |
| NaOH (VWR Chemicals) (g/L) | 160 |

### 2.3. Experimental

Both the dry and tribocorrosion reciprocating sliding tests were performed in a Bruker's UMT TriboLab™ equipped with a 3-electrode tribocorrosion cell using a Pt wire as the counter electrode, and an Ag/AgCl/KCl (saturated) electrode as the reference electrode. The tribometer performs the mechanical loading between the sample and the alumina ball, employed as a counterpart during dry and in corrosive solution test. The continuous measurement of the friction force (Fx) and normal force (Fz) as a function of time provided the value of the coefficient of friction (COF). A Bio-logic SP50 Potentiostat (Seyssinet-France) recorded the open circuit potential (OCP) evolution before, during and after sliding.

The experimental parameters of the two tests are presented in Table 2. The normal load was set to 2 N giving a maximum contact Hertzian pressure of approximately 1.1 GPa.

**Table 2.** Parameters of tribocorrosion and dry sliding tests.

| Parameter \ Test | Tribocorrosion Test | | | Dry Sliding Test |
|---|---|---|---|---|
| Solution | 0.1 M NaCl | | | - |
| Surface area | 1 cm$^2$ | | | - |
| Test duration (min) | Before sliding 20 | During sliding 10 | After sliding 20 | 10 |
| Counter body | Al$_2$O$_3$ | | | Al$_2$O$_3$, φ = 4.8 mm |
| Wear length (mm) | 4 | | | 4 |
| Sliding time (min) | 10 | | | 10 |
| Sliding speed (cm/s) | 2 | | | 2 |
| Load (N) | 2 | | | 2 |

The worn surfaces after the tests were analyzed using an Hirox 3D-digital microscope (Limonest, France) and a Hitachi SU8020 scanning electron microscope (SEM) (Tokyo, Japan) coupled with energy-dispersive X-ray spectroscopy (Thermo Fisher Scientific, Waltham, MA, USA). Surface topography data were obtained with a three-dimensional non-contact optical profilometer (Contour GT, Bruker, Billerica, MA, USA) to obtain additional information on the mechanism of wear under both dry and wet test.

## 3. Results and Discussions

### 3.1. Evolution Open Circuit Potential (OCP) and COF of ENB Coating during the Tests

The OCP potential evolution of the ENB coating obtained from the tribocorrosion test is shown in Figure 1. As can be observed, the OCP value is stabilized around −0.31 V before sliding.

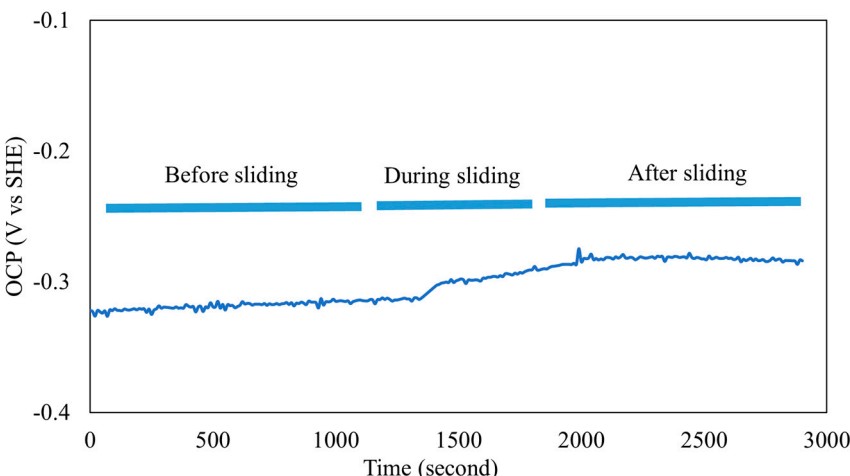

**Figure 1.** Evolution of OCP measured on ENB coating: before, during and after sliding test.

Once the sliding started, the OCP value slightly increased in the anodic direction by 0.03 V. This small gradual increase is attributed to the action of the abrasive $Al_2O_3$ ball producing a removal of the coating accompanied with the modification of the contact conditions, kinetics of the corrosion process in the track and offering a new surface. At this point, it is important to stress out that in its as-deposited condition the coating is not prone to form a passive film in NaCl solution.

Lopez-Ortega et al. [28] have determined that in the case of active metallic materials, the wear track exhibited a higher positive potential than the unworn area, which lead to the formation of a galvanic coupling between the worn and unworn surfaces, resulting in a wear-accelerated corrosion of the latter.

As could be observed, the OCP values in the present case were not significantly affected by loading or unloading of the coated system. However, the slight increase of the OCP potential in the anodic direction from −0.31 V to −0.28 V, indicates that a small galvanic coupling was formed between the wear track material and the unworn material outside it. This implies that the coating outside the wear track corrodes at a higher rate and explains the change of coating morphology, which becomes smoother. When the sliding was stopped and the load was released, the OCP returned to a slightly higher value (by 0.03 V) than the one corresponding to the start of the tribocorrosion test.

Therefore, the total loss of material during the OCP test in the 0.1 M NaCl solution is the result of three different contributions: the material loss due to the wear process, the one due to corrosion in the track and the one due to accelerated corrosion in the unworn areas.

The results obtained in the present investigation are different from those reported by Vitry and Bonin [21] when they studied the tribocorrosion performance of the ENB coatings containing Pb as a stabilizer. They reported a constant OCP value across all of the OCP tribocorrosion tests for an ENB coating thickness around 25 μm. However, when the coatings had a lower thickness of 15 μm, they showed that decrease in the OCP potential occurred during the tribocorrosion test a when the sliding started, accompanied by the detachment of the. Consequently, the coating thickness is of utmost importance for tribocorrosion.

Nevertheless, the value of OCP for the ENB coatings produced without the addition of stabilizer exhibited a smaller value of −0.31 V in 0.1 M NaCl solution than those produced with addition of lead as stabilizer, whose value was of 0.39 V, indicating that the former one had a lower tendency to corrode in the same solution. In fact, potentiodynamic polarization experiments on both types of coatings have shown that the $E_{corr}$ of the ENB deposit was more positive than the one of the ENB-Pb deposit [23,26].

Figure 2 shows the comparison of the COF evolution of the ENB coating during reciprocating sliding wear with and without corrosive solution. It was observed that the COF during the tribocorrosion test is lower (0.2) in comparison to that obtained in the dry

sliding test (0.36), which is in agreement with the findings of several researchers [14,15,18], a fact that could be attributed to the lubricating action of the electrolyte, carrying away the debris generated by the contact of triboelements. It was found that a steady state of the COF can be achieved during the tribocorrosion test, while the constant production of debris and their presence in the contact implied an increase in the COF with distance during the dry wear test.

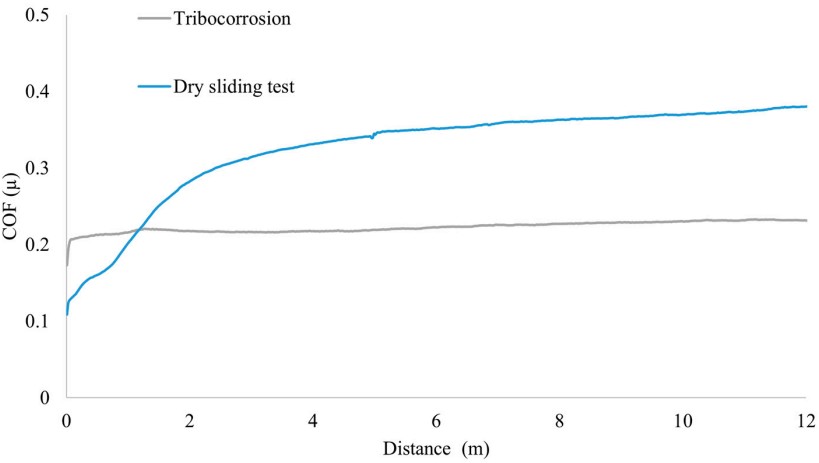

**Figure 2.** COF of the ENB coatings obtained from the dry sliding and tribocorrosion test.

*3.2. Wear Mechanisms and Morphologies of the Counterparts*

3.2.1. Dry Sliding Tests

The worn surfaces of the coatings tested during reciprocating sliding and the surface of the counterpart are shown in the SEM micrographs presented in Figure 3a,b, respectively. In Figure 3a, longitudinal and deep grooves can be observed in the middle of the wear track zone due to the plowing action of the alumina ball, indicative of an abrasive wear mechanism [29,30]. At the side of the wear track the presence of ridges due to the accumulation of debris could also be observed.

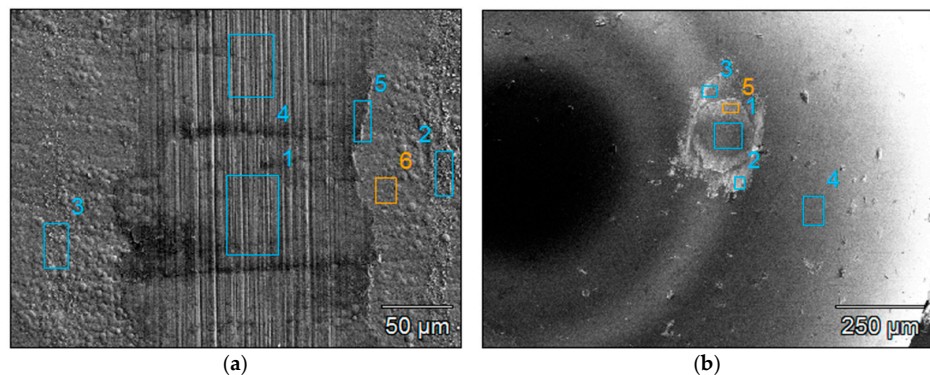

(a)          (b)

**Figure 3.** (**a**) Surface morphology of the worn coating. (**b**) Surface of the ball after the dry sliding test.

Different areas on the SEM micrograph of the wear track were analyzed by EDS and their compositions are reported in Table 3a, corroborating this mechanism. The presence of oxygen indicates that oxidation of Ni took place during sliding due to local temperature increase, indicating that debris will be constituted of a mixture of alumina and nickel oxides.

The surface morphology of the worn ball is observed in Figure 3b. In this case, the EDS analysis (Table 3b) revealed the presence of aluminum, oxygen, nickel and carbon (from the process of making the ball conductive). There is a significant amount of nickel in the oxidized debris attached to the ball, which is a sign of the adhesive wear mechanism

promoted by the difference in hardness between the counterparts, as well as the chemical compatibility between oxides.

**Table 3.** (**a**) EDS analysis of the coating surface after the dry sliding test showing the chemical composition (weight percentage) of different points outside and inside the wear track. (**b**) EDS analysis of the alumina counterpart surface after the dry sliding test showing the chemical composition (weight percentage) of the adhered material.

| (a) | | | | | |
|---|---|---|---|---|---|
| Element / Region | **C** | **O** | **Al** | **Fe** | **Ni** |
| 1 | 0.8 | 8.5 | 1.1 | 1.1 | 88.5 |
| 2 | 2.3 | 6.5 | 0.7 | 0.8 | 89.7 |
| 3 | 1.9 | 3.5 | | 0.9 | 93.7 |
| 4 | 0.7 | 6.2 | 0.7 | 1.1 | 91.3 |
| 5 | 1.2 | 13.0 | 1.4 | 0.9 | 83.5 |
| 6 | 0.9 | | | 1.3 | 97.8 |

| (b) | | | | |
|---|---|---|---|---|
| Element / Region | **C** | **O** | **Al** | **Ni** |
| 1 | 9.3 | 69.1 | 21.6 | |
| 2 | 13.8 | 40.5 | 8.0 | 37.7 |
| 3 | 7.8 | 50.8 | 17.3 | 24.1 |
| 4 | 15.6 | 61.7 | 22.8 | |
| 5 | 8.4 | 47.9 | 13.9 | 29.8 |

### 3.2.2. Tribocorrosion Tests

Wear tracks on the coating surface and ball after the tribocorrosion tests are shown in the SEM micrographs presented in Figure 4. Like in the case of the dry sliding tests, longitudinal micro-grooves are observed (see Figure 4a). Debris particles are not found on the wear track, since they were swept away during the reciprocating movement of the sample in contact with the electrolyte. In the wear track, material loss is also observed, indicating an abrasive wear mechanism. A small percentage of Al was found in the wear track (see Table 4a) as a consequence of a slight abrasive mechanism exhibited by the alumina ball in contact with the coating. The amount of oxygen (Table 4a) found in the wear track during the tribocorrosion test is much less than that found in the wear track formed after the dry sliding test (Table 3a), since the presence of the electrolyte, as expected, decreased the oxidation during the tribocontact by hindering local heating.

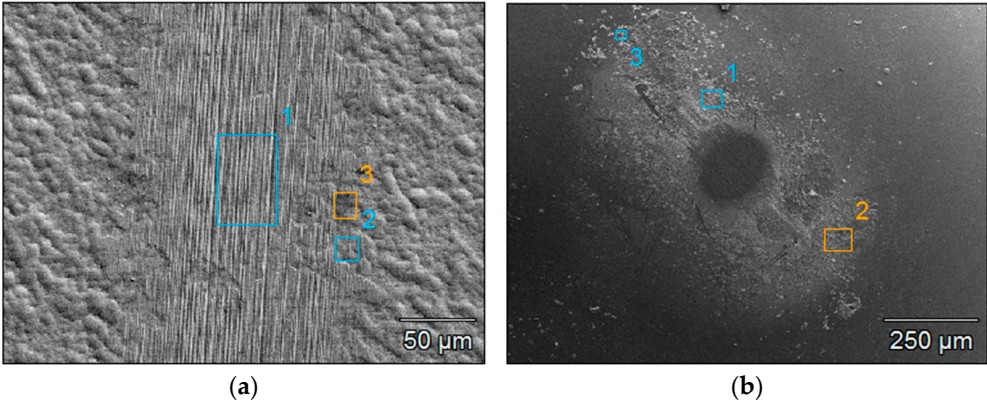

| (**a**) | (**b**) |

**Figure 4.** (**a**) Surface morphology of the worn coating. (**b**) Surface of the ball after the tribocorrosion test.

**Table 4.** (**a**) EDS analysis of the coating surface after the tribocorrosion test showing the chemical composition (weight percentage) of different points outside and inside the wear track. (**b**) EDS analysis of the alumina counterpart surface after the tribocorrosion test showing the chemical composition (weight percentage) of the adhered material.

| (a) | | | | | | | |
|---|---|---|---|---|---|---|---|
| Region \ Element | C | O | Na | Al | Si | Cl | Ni |
| 1 | 0.9 | 3.8 | | 1.0 | | 0.3 | 94.0 |
| 2 | 1.4 | 4.3 | | 1.2 | | 0.2 | 93.0 |
| 3 | 2.4 | 9.8 | 1.0 | 1.9 | 0.2 | 0.3 | 84.4 |

| (b) | | | | | | |
|---|---|---|---|---|---|---|
| Region \ Element | C | O | Na | Al | Cl | Ni |
| 1 | 6.7 | 37.2 | 4.3 | 22.2 | 4.7 | 24.9 |
| 2 | 6.4 | 16.4 | 26.6 | 10.8 | 31.7 | 8.0 |
| 3 | 4.4 | 37.7 | 12.3 | 26.7 | 14.1 | 4.8 |

It is interesting to observe in these micrographs the difference between the morphology of the coating outside the wear track after dry sliding (Figure 3a) and after the tribocorrosion test (Figure 4a). In the latter case, due to corrosion, the grains are revealed and exhibit a much smoother morphology.

The presence of a small amount of nickel on the ball surface is a sign that some adhesive wear (see Table 4b) took place. Table 4 also indicates the presence of Na and Cl, originated from the electrolyte.

### 3.3. Profilometry Analysis of the Wear Tracks

The profilometry analysis provides some additional information and corroborates the previous findings regarding the coatings wear mechanisms during both tribotests. Figure 5 illustrates a representative profilometry image corresponding to the wear track of the coated sample tested in air (dry conditions).

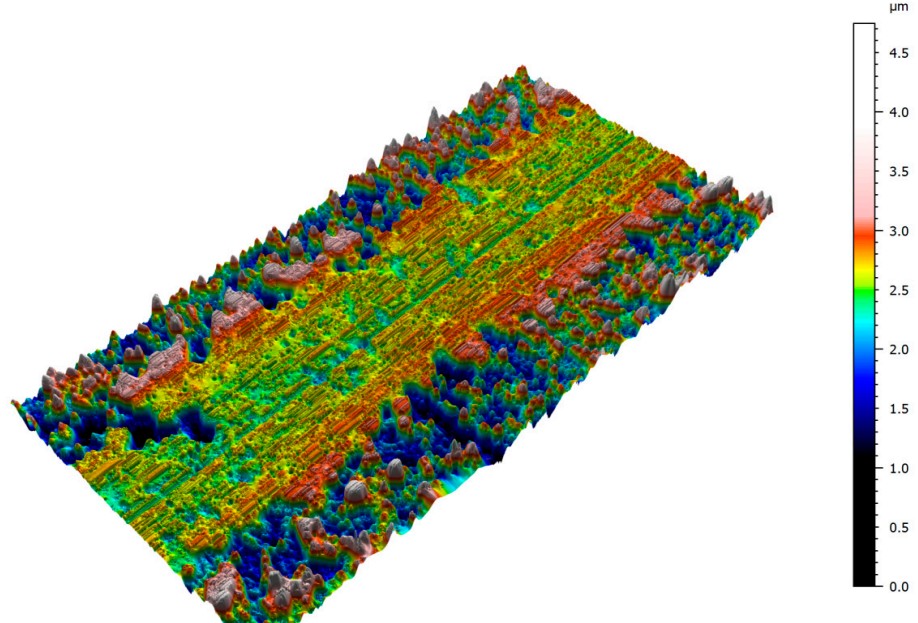

**Figure 5.** 3D-profilometry image taken close to the middle of the wear track, corresponding to a sample tested in air (dry conditions).

The image analysis indicates that the passage of the ball along the track leads to the smoothening of its surface as compared to the morphology of the coating surface outside it, as well as the crushing of the asperities. The oxidized debris material, which is detached from the asperities, fills in the holes and also gives rise to the presence of ridges on each side of the track. For comparison, Figure 6 illustrates a similar profilometry image from the wear track of a sample tested under tribocorrosion conditions. A close examination of the track indicates that it is similar to the zone observed on the sample tested under dry conditions, although some differences can be found.

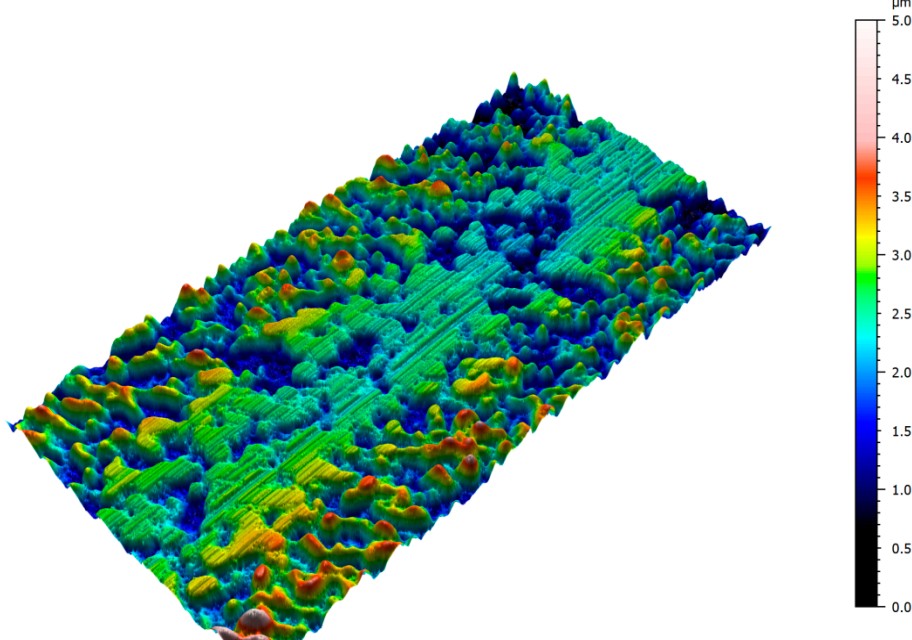

**Figure 6.** 3D-profilometry image corresponding to a sample tested under tribocorrosion conditions.

On the topographic image illustrated in Figure 6, it can be seen that the peaks are smoother than those observed in the dry test, as a consequence of their partial dissolution induced by corrosion. However, in this case, the holes on the track cannot be filled in since the solution hinders the deposition of the debris detached from the asperities. On the contrary, this material is swept away by the solution and, therefore, under these conditions the ridging previously observed cannot be built on the track sides.

In order to compute the wear area at particular locations along the wear tracks, three different profiles were taken on the tracks after testing both in air and under corrosive conditions. Figures 7 and 8 illustrate representative examples of the profiles obtained in each case. As indicated above in relation to Figure 5, it can be clearly observed that at the center of the track the profile tends be quite smooth, with changes in height of less than 1 μm in magnitude, as a consequence of the filling up of the holes with debris material. The computation of the wear area was conducted taking into account the actual track width, as observed in Figure 3a, whose mean value is of approximately 180 μm$^2$.

In the case of the samples tested under tribocorrosion conditions (Figure 8), in agreement with the observations made in relation to Figure 6, the profile clearly illustrates that the track has not been smoothed by the passage of the ball, which leads to changes in the height profile between of approximately 1.5 μm in magnitude. As explained before, under these conditions, the debris material is removed from the track by the corrosive solution and cannot be deposited on the holes present on its surface, which explains such a variation. Despite the significant differences between the two types of profiles, an attempt was made to determine the wear area at the different locations where the profiles were taken, following a simple computational algorithm based on the fit of a spline with cubic ends to each profile, for interpolating the height values within the track width. For this

purpose, the track width was divided in 1500 intervals and the interpolated function was integrated numerically for determining the area below the wear profile.

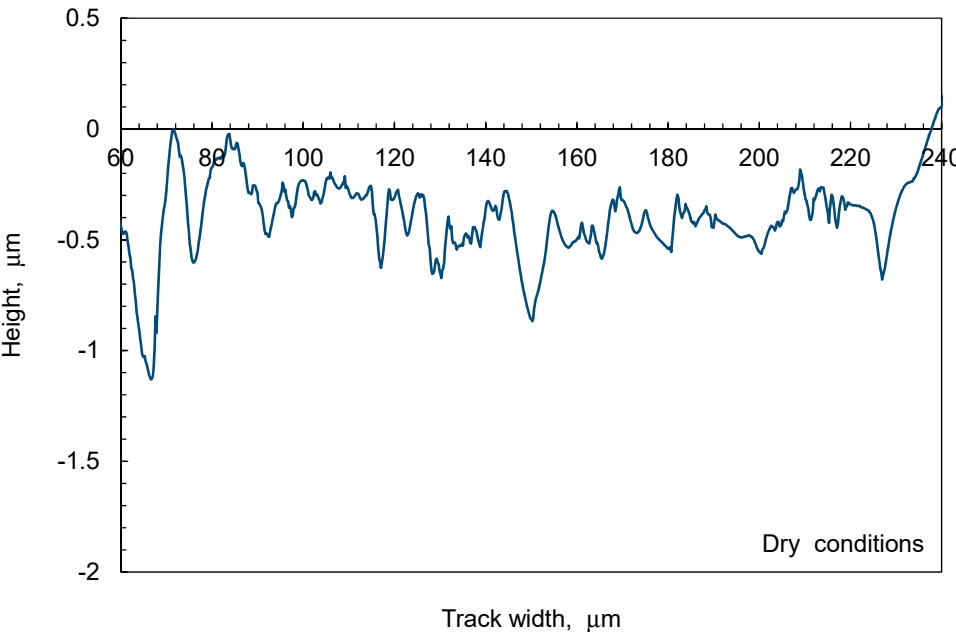

**Figure 7.** Typical wear profile obtained along the track of a sample tested under dry conditions. The wear track has a mean width of 180 $\mu m^2$.

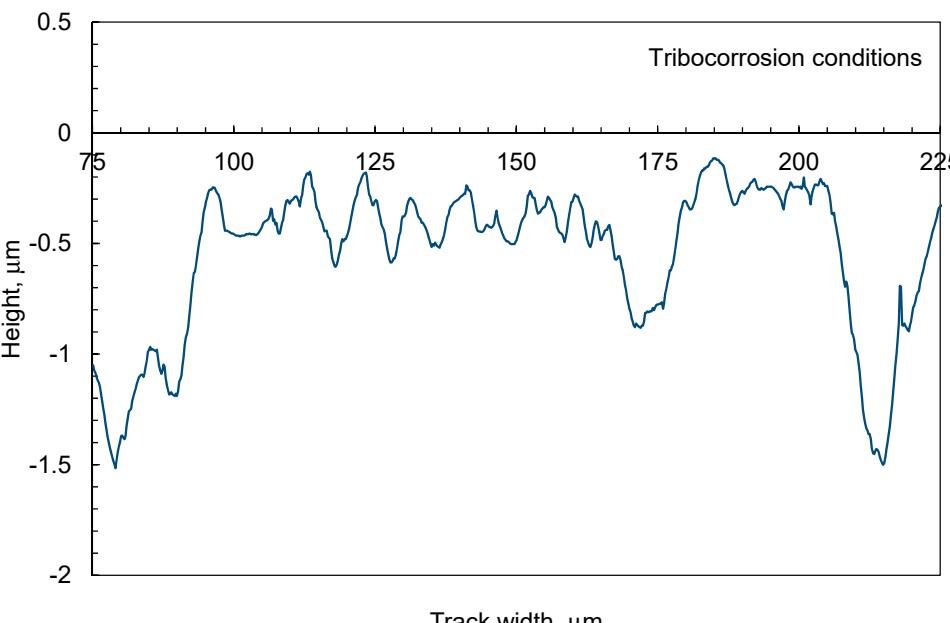

**Figure 8.** Typical wear profile obtained along the track of a sample tested under tribocorrosion conditions. The wear track has a mean width of approximately 150 $\mu m^2$.

The results of this analysis indicated that the wear rates areas under the profiles were of 86 ± 15 and 56 ± 25 $\mu m^2$ for the dry and tribocorrosion tests, respectively. Taken into account that, during the tests conducted under tribocorrosion conditions, the holes on the track cannot be filled in by the detached debris and given the wider wear tracks observed for the dry tests, as can be seen by comparing Figures 3a and 4a. Therefore, it can be concluded that, during the wear tests conducted in air, the coating undergoes a more

severe wear than in the case of the tests carried out in the corrosive solution, a fact which corroborates the evolution of the friction coefficients during both tests.

## 4. Conclusions

Lead-free ENB coatings were produced in the stabilizer-free bath. Their tribocorrosion behaviour was investigated. It was found that a lower COF (0.2) observed in the tribocorrosion test due to the lubrication effect of the electrolyte. In the case of dry sliding wear test, the COF is 0.36. The electrolyte swept away the debris, which resulted in reaching a steady COF at an earlier time.

It was shown that the ENB coating did not show any passivation. An increase in the OCP was also shown, which resulted in galvanic coupling between the worn area and the unworn area. Analyzing the wear track on the coatings showed that an abrasive corrosion-wear mechanism was predominant in tribocorrosion test. However, in the case of the dry sliding test, abrasive and tribo-oxidation wear mechanisms were observed. The main reason for these results was the effect of the electrolyte that hindered oxidation between the counterpart and coating surface.

The total loss of material in the wear track during the sliding tests in the corrosive solution is the sum of three components: material loss due to wear, material loss due to corrosion in track and the material loss produced by the accelerated corrosion in the outer part of the track. Profilometry analysis exhibited that the wear occurred during the sliding test ($86 \pm 15$) was more severe than the one occurred during the tribocorrosion test ($56 \pm 25$), attributing the lubrication effect of the electrolyte.

**Author Contributions:** Conceptualization, M.Y. and V.V.; methodology, M.H.S.; software, A.M.; validation, V.V., A.M. and M.H.S.; formal analysis, V.V.; investigation, M.Y.; resources, M.Y.; data curation, M.H.S. and A.M.; writing—original draft preparation, M.Y.; writing—review and editing, V.V., M.H.S. and A.M.; visualization, M.Y.; supervision, M.H.S.; project administration, V.V.; funding acquisition, V.V. and A.M. All authors have read and agreed to the published version of the manuscript.

**Funding:** This study was supported by the INTERREG VA program and European Regional Development Fund (FEDER) in the framework of the AltCtrlTrans project.

**Institutional Review Board Statement:** Not applicable.

**Informed Consent Statement:** Not applicable.

**Data Availability Statement:** Data are contained within the article.

**Acknowledgments:** The authors gratefully acknowledge INTERREG for funding. The authors also would like to thank Yoann Paint from Materia Nova for his help with the analysis of the samples by SEM. Morphlogical datasets and related data analysis are provided by the Morphomeca platform and are managed by LAMIH UMR CNRS 8201, National Institute of Applied Science (INSA), Polytechnic University of Hauts-de-France, Le Mont Houy, Valenciennes, France.

**Conflicts of Interest:** The authors declare no conflict of interest.

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
