# Peer review of "Replacing Toxic Hard Chrome Coatings: Exploring the Tribocorrosion Behaviour of Electroless Nickel-Boron Coatings"

_coatings, doi:10.3390/coatings13122046_

Round 1

Reviewer 1 Report

Comments and Suggestions for Authors

After going through the paper titled "Replacing Toxic Hard Chrome Coatings: Exploring the Tribocorrosion Properties of Electroless Nickel-Boron Coatings" I would suggest the following changes:

1. Is tribocorrosion a property or system response? This should be checked in the title. It may be "tribocorrosion behavior". This has been correctly written in the abstract.

2. Some important quantified gains should be given in the abstract. For example how much percent the wear rate improved?

3. While emphasizing on the fact that the wear and corrosion has been investigated separately by several researchers, the suitability of the coatings in corrosive environment should be discussed in the Introduction. There are different Ni-B variants. The lead stabilized coatings have columnar structure and have moderate corrosion resistance. While the binary and poly alloys of Ni-B (lead stabilized or lead free) do not have such growths. So the corrosion resistance should be discussed in the literature. 

4. Line 96, how was the thickness determined? Through cross-section or mass loss or in a previous study? In that case reference should be given.

5. Lines 98-99, how was the coating structure determined? Also what was the basis for selection of the bath paramaters? There is lack of adequate characterization of the coatings post deposition. Are they referred from the authors previous works? Then that should be mentioned.

6.  In Table 2, I think the reciprocating frequency is given instead of speed.

7. Either for the dry sliding case or under the corrosive media, the wear rate is not mentioned. Since the surface is quite irregular, the mass loss of the coatings may be reported and compared.

8. Lastly, the conclusion should be crisp. Here also no quantified gains are mentioned. It is more like an abstract. 

9. The improvement in wear rate or COF in terms of percentage (worn area or mass loss or wear rate) should be given.

The paper thus requires minor revisions in terms of its characterization and elaborate literature review. The literature review should be expanded since a significant work has been reported recently to justify the context of the present work. In fact Ni-B-W coatings have been also reported as a substitute to hard chrome. There is Ni-B-Mo coatings also. Their corrosion resistance is also reported in the literature. Thus what is lacking in these new works should be highlighted to justify the tribo-corrosion damage evaluation. Else the paper is well written in terms of characterization of worn surface.

Author Response

Dear Reviewer, 

I am quite appreciated for your comments? All of your comments were addressed in the article. 

  1. The title was changed the new title is "Replacing Toxic Hard Chrome Coatings: Exploring the Tribocorrosion Behaviour of Electroless Nickel-Boron Coatings" In the article as well, it was shown as "tribocorrosion behaviour".
  2. In the abstract, the wear area obtained after each tests was introduced.
  3. Corrosion resistance of ENB-Pb (lead stabilized) and ENB (lead free) is shown in the results and discussion section (line 177-182).
  4. The thickness was measured on cross section via optical microscopy, this information is written in the section 2.2 (line 119).
  5. The structure of the coating was investigated in the previous study.  The reference is mentioned. (Line 121).
  6. The table 2 was modified. You can see the change.
  7. The wear area occurred after the test is shown at line 300. As the other parameters (sliding distance, sliding lenght etc) are the same, only wear area is mentioned for comparison. 
  8. Conclusion was modified, you can see the change (line 311-328). 
  9. COF data is given in the conclusion (line 312-315). 

I hope our answers address your comments, otherwise we would be happy to receive your feedback. 

Reviewer 2 Report

Comments and Suggestions for Authors

Yunacti et al investigated the tribocorrosion behaviour of prepared electroless nickel-boron coating on mild steel. The experimental processes were described in detail. The surfaces of the samples were also characterized well. The experiments showed potential applications of the EBN coatings to protect the steels. The manuscript is written properly. The text is in the scope of this Journal. Thus, I’d like suggest acceptance of this manuscript for publication after some minor improvements.

1). The term ‘COF’ (line 20) was used in the Abstract with definition, which may cause difficult for readers.

2). Term ‘OCP’ in line 60 should be defined first.

Comments on the Quality of English Language

It is OK.

Author Response

Dear, 

I would like to thank you for your valuable comments.

In the abstract, both COF and OCP were defined. 

Reviewer 3 Report

Comments and Suggestions for Authors

This manuscript's novelty is so low. Many errors inside the manuscript.

The introduction is very simple. No novelty is highlighted. Many English mistake. I don’t find any new findings for this manuscript.

The conclusion section needs to be rewritten. Don’t write point-wise. Make it into a paragraph. Include the major findings.

Line 308/309 remove these words for research articles with several authors, a short paragraph specifying their individual contributions must be provided. The following statements should be used.

EDX mapping figure is necessary.

All figure axis thik marks are missing. Figures resolution is low.

Section 3 should be written as Results and Discussion.

Many paragraphs are unnecessary; please rearrange it. Avoid many paragraphs.

References are too less.

Discussion is insufficient.

Comments on the Quality of English Language

English very difficult to understand/incomprehensible

Author Response

Dear Reviewer, 

First of all, I would like mention my appreciation for your valuable comments. The article was modified based on your comments. 

The novelty of the study was explained in the introduction (line 89 to 93).

The introduction section was modified and the change can be seen. 

The conclusion section was re-written. The COF and wear area which are the major findings were mentioned. 

The section of author contributions was modified. And the sentence " for research articles with several authors, a short paragraph specifying their individual contributions must be provided" was removed. Line 348.

Instead of EDX mapping, the EDX analysis on different parts of the wear track and the ball are shown in Table 4 and Table 5. 

Figures resolution were enhanced. 

Section 3 was changed. It is shown as results and discussions. 

Paragraphs were rearranged. Section 3.1 and 3.2 were combined, and then unnecessary paragraphs were removed. 

More references were added. Now there is 30 references. 

Results and discussions section was modified and more references were added for disscusions. 

I hope my answers will address your comments, otherwise I am at your disposal. 

Round 2

Reviewer 3 Report

Comments and Suggestions for Authors

Check the typo errors during proofreading.

Comments on the Quality of English Language

mimor checking is needed.